# Roles of Scaling and Instruction Tuning in Language Perception: Model vs. Human Attention

**Changjiang Gao[1]**    **Shujian Huang[1]***    **Jixing Li[2]**    **Jiajun Chen[1]**

[1]National Key Laboratory for Novel Software Technology, Nanjing University
[2]Department of Linguistics and Translation, City University of Hong Kong
gaocj@smail.nju.edu.cn    {huangsj, chenjj}@nju.edu.cn
jixingli@cityu.edu.hk

## Abstract

Recent large language models (LLMs) have revealed strong abilities to understand natural language. Since most of them share the same basic structure, i.e. the transformer block, possible contributors to their success in the training process are scaling and instruction tuning. However, how these factors affect the models' language perception is unclear. This work compares the self-attention of several existing LLMs (LLaMA, Alpaca and Vicuna) in different sizes (7B, 13B, 30B, 65B), together with eye saccade, an aspect of human reading attention, to assess the effect of scaling and instruction tuning on language perception. Results show that scaling enhances the human resemblance and improves the effective attention by reducing the trivial pattern reliance, while instruction tuning does not. However, instruction tuning significantly enhances the models' sensitivity to instructions. We also find that current LLMs are consistently closer to non-native than native speakers in attention, suggesting a suboptimal language perception of all models. Our code and data used in the analysis is available on GitHub.

## 1 Introduction

Large language models (LLMs), e.g., GPT-4, Chat-GPT and Vicuna, have shown nearly human-level understanding of text inputs, indicating better human-like language perception compared with their predecessors such as BERT (Devlin et al., 2019) and GPT-2 (Radford et al., 2019). However, the mechanism behind such improvement is largely unknown. One way to interpret such mechanism is comparing model computation processes to human data. For example, prior work compared language models (mainly GPT-2) with human neuroimaging data during language comprehension, suggesting that models with higher human resemblance also perform better in NLP tasks. (Hasson

---

* Corresponding author

et al., 2020; Schrimpf et al., 2021). However, given recent breakthroughs in LLMs , it remains to be tested whether the newest LLMs still align with human perception data. Based on the training pipeline of current LLMs, there are two possible sources for the potential change: the scaled model/data size and the aligning process after pretraining, such as instruction tuning (Ouyang et al., 2022; Wei et al., 2022; Sanh et al., 2022). This paper aims to provide a further understanding of LLMs' mechanism by investigating the roles of these two factors in language understanding.

In this paper, we analyze self-attention as a channel to the language perception ablility of LLMs, because it is the key mechanism of transformer language models, and it naturally resembles the human attention mechanism in language perceiving (Merkx and Frank, 2021). The attention patterns under investigation includes those from trending open-sourced LLMs with different sizes, i.e., 7B, 13B, 30B, 65B, and different training stages, i.e., pretrained (LLaMA) and instruction-tuned (Alpaca and Vicuna) models. Given these models are all trained mainly with English data, we also compare the attention patterns of LLMs with human saccade (a form of eye-movement) data, including both native speakers (L1) and non-native learners (L2) of English (Li et al., 2019) to see if the models correlates more with L1 subjects.

Our analysis is three-fold. First, we compare the general attention distributions of LLMs to measure the impact of these two varying factors on the model (§3.1). Second, we perform linear regressions (LRs) between the model self-attention and human saccade matrices to see how human-like they are, and how the resemblance is affected by the two factors (§3.2). Third, we analyze the relation between the given attention and some trivial patterns, revealing the difference between different levels of language perception abilities (§3.3). The main findings are:

- Scaling significantly affects the general attention distribution on plain text, while instruction tuning has limited effect. However, instruction tuning enhances the model's sensitivity to instructions (§5.1).

- Higher human resemblance is significantly correlated to better language modeling. Scaling improves the resemblance by scaling law (Henighan et al., 2020), while instruction tuning reduces it. All models have higher resemblance to L2 rather than to L1, suggesting further room for the improvement in language perception (§5.2).

- L2 saccade has higher dependency on trivial attention patterns than L1, indicating a more rigid way of perception. Scaling significantly lowers LLMs' reliance on trivial attention patterns, while instruction tuning does not (§5.3).

## 2 Related Work

**Self-attention analysis** is a common way to interpret transformer models. Several studies has shown that it is correlated to linguistic information, such as syntactic or semantic structures (Mareček and Rosa, 2018; Raganato and Tiedemann, 2018; Voita et al., 2019; Clark et al., 2019). There are also some attempts and debates to explain the models' prediction with attention (Jain and Wallace, 2019; Wiegreffe and Pinter, 2019; Serrano and Smith, 2019; Vashishth et al., 2019). Different from them, we use self-attention patterns as a measurement of the language perception abilities of LLMs. Also, some new ways to interpret model self-attention have been proposed, such as Attention Flow (Abnar and Zuidema, 2020), vector norm (Kobayashi et al., 2020, 2021), ALTI (Ferrando et al., 2022), Value Zeroing (Mohebbi et al., 2023), and logit analysis (Ferrando et al., 2023). These methods can raise the interpretability of model attention w.r.t. linguistic structural information, but are not tested with human eye-tracking data. Also, some of them suffer from huge computational cost. Instead, we choose to use the raw attention scores, as it is produced in the models' original computational processes, and they already correlate the human data, demonstrating their interpretability.

**Instruction tuned LLMs** achieve better performances in performing tasks (Ouyang et al., 2022; Wei et al., 2022; Sanh et al., 2022). It seems that the LLMs are better at understanding human instructions. However, there is little discussion on how the instruction tuning process affects the language perception.

**Eye-movement in the human reading process** has drawn attention in the computational linguistic field. It is proposed that scanpaths, i.e. tracks of eye-movement, reveals characteristics of the parsing processes and relates to working memory during reading (von der Malsburg and Vasishth, 2013), and the irregularity of scanpaths is affected by sentence structures and age (von der Malsburg et al., 2015). Such work provides theoretical ground for our analysis on saccade data.

**Joint study of language models and human eye-movement** is also receiving growing interest. Prior studies have shown the correlation between language processing of human and artificial neural networks (Kell et al., 2018), especially pretrained language models such as GPT-2 (Caucheteux et al., 2021, 2022; Schrimpf et al., 2021; Lamarre et al., 2022). Especially, Oh and Schuler (2023a,b) explored the fit of model surprisal to human reading times, noting the effect of models' sequential memorization and dataset size. Hollenstein and Beinborn (2021) studied the correlation between relative word importance derived from models and humans, and Morger et al. (2022) extended that research to a multilingual setting, as well as analysis on model attention. Some studies also use human attention as supervision to assist specific tasks (Sood et al., 2020; Bansal et al., 2023), or use sequential models to predict eye-tracking data (Chersoni et al., 2022). Different from them, we use human attention as references for the language perception abilities.

## 3 Methods

To show the effect of scaling and instruction tuning on different models, we compare the self-attention scores of different LLMs given the same input, by viewing model attention as probability distributions and calculating the general attention divergence based on Jensen-Shannon divergence.

To analyze model self-attention, we take the human saccade as a reference, and design a human resemblance metric based on linear regression scores.

In addition, we select several trivial, context-free attention patterns, and design a trivial pattern reliance metric to help demonstrate the difference

between models and human subject groups.

## 3.1 General Attention Divergence

We assesses the mean Jensen-Shannon (J-S) divergence between attention from different models on all sentences. For two attention matrices $A,\ B \in \mathbb{R}^{n_{\text{word}} \times n_{\text{word}}}$, the J-S divergence is:

$$D_{\text{JS}}(A, B) = \frac{1}{2} \sum_{i=1}^{n_{\text{word}}} [D_{\text{KL}}(A_i \| B_i) \\ + D_{\text{KL}}(B_i \| A_i)]$$

where $D_{\text{KL}}$ is the Kullback–Leibler (K-L) divergence, $A_i$ and $B_i$ are the $i$-th rows in the two matrices. Note the attention matrices are re-normalized to keep the sum of every row to 1.0. We calculate the J-S divergence between heads-averaged attention scores per sentence, and demonstrate the mean values. For models with the same number of layers, the divergence is calculated layerwise; For models with different numbers of layers, we divide their layers into four quarters and calculate the divergence quarter-wise.

To identify high and low divergence, a reference value is required. We propose to use the attention divergence between two models that share the same structure, size, and training data, but have small difference in training strategies as the reference. A divergence value higher than the reference suggests relatively large change attention, while one lower than that indicates small (but not necessarily negligible) change.

**Divergence by Scaling and Instruction Tuning** Models with different scales or training stages are compared with the above divergence to show the effect of different factors.

**Sensitivity to Instructions** To evaluate the effect of instruction tuning, we measure a given model's sensitivity to instructions by the same general attention divergence metric, but on different input, i.e., plain input and input with instructions. If the divergence between them is significantly higher than reference, we say the model has high sensitivity to instructions.

To construct instructed text, we attach two prefixes to plain text sentences: "Please translate this sentence into German:", and "Please paraphrase this sentence:". They are within the reported capacity of most LLMs, and require no extra information to follow. Then, we collect model attention on

the two corpora within the original sentence spans (re-normalized) to represent non-instruction and instruction scenarios. Also, as a control group, we attached a noise prefix made of 5 randomly sampled English words "Cigarette first steel convenience champion.", and calculate the divergence between prefixed and plain text to account for the effect of a meaningless prefix.

## 3.2 Human Resemblance

### 3.2.1 Human Resemblance Calculation

Given the model self-attention and the human saccade data in matrix form for each sentence, we first extract the lower-triangle parts of the matrices (marking right-to-left attendance), because prior studies have found that such eye-movements in reading are related to working memory efficiency (Walczyk and Taylor, 1996) and parsing strategies (von der Malsburg and Vasishth, 2013), and the auto-regressive LLMs only have right-to-left self-attention. Then, we flatten the extracted parts of each sentence attention matrix, and concatenate them to get attention vectors $\{v_{\text{human}}^i\}$ for each subject $i$, and $\{v_{\text{model}}^{j,k}\}$ for head $k$ in model layer $j$. We then stack $\{v_{\text{model}}^{j,k}\}$ along the attention heads and get a matrix $\{V_{\text{model}}^j\}$ for the $j$-th layer. The LR for model layer $j$ and subject $i$ is defined as:

$$\arg \min_{w,b} \left\| V_{\text{model}}^{j\top} w + b - v_{\text{human}}^i \right\|_2^2$$

This means we take the linear combination of attention scores from all heads in a layer to predict the human saccade. Note that the model attention here are not re-normalized, in order to preserve the relative weight w.r.t. the \<s\> token. The regression scores $R_{i,j}^2$ (ranging from 0 to 1, larger value means better regression) are averaged over subjects to represent the human resemblance of layer $j$, and the highest layer score is considered as the regression score of the whole model, $R_{\text{model}}^2$.

Because the human saccade pattern varies across individuals, there exists a natural upper bound for human resemblance of models, i.e., the human-human resemblance, or the inter-subject correlation. To calculate this correlation within the two language groups, L1 and L2, we take the group mean to regress over each individual data independently, and use the average regression score $R_{\text{inter}}^2$ as the inter-subject correlation for that group. The final human resemblance score is $R_{\text{model}}^2 / R_{\text{inter}}^2$, ranging from 0 to 100%.

### 3.2.2 Human Resemblance vs. Next-Token Prediction

A previous study finds a positive correlation between the next-token predication (NTP) performance and the "Brain Score" of neural networks (Schrimpf et al., 2021), which supports the intuition that the more human-like models are, the better generation they output. To test this finding in our setting, we calculated the per-token prediction loss (the negative log likelihood loss normalized by sequence lengths) of the models employed in this study on the Reading Brain dataset, and calculate the Pearson's Correlation between their prediction loss and max layerwise human resemblance scores.

### 3.3 Trivial Pattern Reliance

#### 3.3.1 Trivial Pattern Reliance of Models

We select three representative trivial patterns of transformer attention: attending to the first word in the sentence, attending to the previous word (Vig and Belinkov, 2019), and self-attending (Clark et al., 2019), and analyze the effect of scaling and instruction tuning on models' reliance on them.

First, we represent the trivial patterns as binary adjacent matrices, where attending relations are marked with 1, others with 0, and use their flattened lower-triangle parts as the independent variables. The model attention are similarly processed to be the dependent variables. Then, an LR is fit between the two, and the regression scores are collected to represent the trivial pattern reliance of each model layer.

#### 3.3.2 Trivial Pattern Reliance of Humans

This research compares the models' resemblance to L1 and L2 humans because intuitively, L1 is better than L2 in language understanding. To test this intuition, we compare their reliance on the three trivial patterns to see whether their attention mode differs. Similar to model attention, the flattened human saccade vectors are used as the dependent variable to fit the LRs with the trivial patterns.

## 4 Experiment Settings

### 4.1 Data

We use the Reading Brain dataset (Li et al., 2019) for both the text input and the human data. The dataset includes 5 English STEM articles. Each article has 29.6±0.68 sentences of 10.33±0.15 words. It also includes human eye-tracking and fMRI data recorded synchronously during self-paced reading

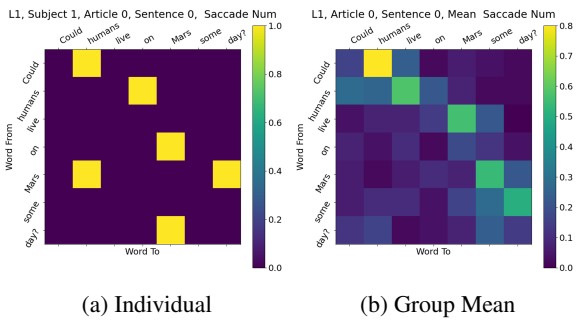

(a) Individual      (b) Group Mean

Figure 1: Examples of individual and group mean saccade of the sentence "Could humans live on Mars some day?".

on the articles. The human subjects are 52 native English speakers (L1) and 56 non-native learners of English (L2). We choose this dataset for the following reasons: (1) The text is presented to the subjects sentence by sentence instead of page by page, making in-sentence reading patterns clearer. (2) It has synchronous eye-tracking and fMRI data, which allows both behavioral- and neural-level analysis. In this article we mainly focus on the behavioral part, but we also have some preliminary results on the fMRI analysis (See Appendix D).

We take the saccade number, i.e. times of eye-movements from a word to another, instead of saccade duration, to represent the human reading attention, because this reduces the effect of other factors such as word length. The saccade data of a human subject for a given sentence is a $n_{\text{word}} \times n_{\text{word}}$ matrix. Figure 1 gives examples of individual and group mean saccades.

### 4.2 LLMs

We employ LLaMA (Touvron et al., 2023) and its instruction-tuned versions, Alpaca (Taori et al., 2023) and Vicuna (Chiang et al., 2023) in this research. We use them because: (1) The model sizes and different instruction-tuning methods covers the majority of current open-sourced LLMs. (2) These models are being widely used and customized globally. Thus, analysis on them is representative and meaningful.

**LLaMA** is a series of pretrained causal language models with parameter sizes 7B, 13B, 30B and 65B, trained on over 1T publicly available text tokens, and reaches state-of-the-art on most LLM benchmarks (Touvron et al., 2023). The training corpus is mainly English. We use all 4 sizes of LLaMA, which covers the sizes of most current

open-sourced LLMs. We also use the 774M GPT-2 Large (Radford et al., 2019) to represent smaller pretrained models in the scaling analysis.

**Alpaca** is fine-tuned from the 7B LLaMA model on 52K English instruction-following demonstrations generated by GPT-3 (Brown et al., 2020) using self-instruct (Wang et al., 2023). To analyze the effect of scaling, we also trained a 13B version of Alpaca using the official data and training strategy of Alpaca. Our 13B Alpaca model scores 43.9 and 46.0 on the MMLU dataset in zero-shot and one-shot setting respectively, proving the soundness of our fine-tuning. (The corresponding scores are 40.9 and 39.2 for the official Alpaca 7B model.[1])

**Vicuna** models are fine-tuned from the 7B and 13B LLaMA models on 70K user-shared conversations with ChatGPT. The data contains instruction and in-context learning samples in multiple languages, thus the Vicuna models can also be viewed as instruction-tuned.

### 4.3 LLMs Attention Collection

To obtain model attention, text is provided to the models sentence by sentence to perform next token predict, and we collect the attention tensors in each layer. This resembles the human self-paced task-free reading process, where only one sentence is visible at a time to the subjects, and the natural human behavior is to predict the next word (Kell et al., 2018; Schrimpf et al., 2021).

Because models use the SentencePiece tokenization (Kudo and Richardson, 2018), but the human saccade data is recorded in words, the attention matrices need to be reshaped. Following Clark et al. (2019) and Manning et al. (2020), we take the sum over the "to" tokens, and the average over the "from" tokens in a split word. Also, the  token is dropped as it is not presented to human subjects.

**Reference of Divergence** We use the attention divergence of Vicuna v0 and Vicuna v1.1 13B as the reference values. The two versions share the same training data and strategy, but have minor difference in separators and loss computation[2]. We use the divergence calculated on 13B Vicunas across all divergence analyses, because it has lower mean and variance than the 7B one, making our comparison fairer and stricter.

---

[1]Based on this GitHub repository for LLM evaluation.
[2]According to notes in the Vicuna GitHub repository.

## 5 Results

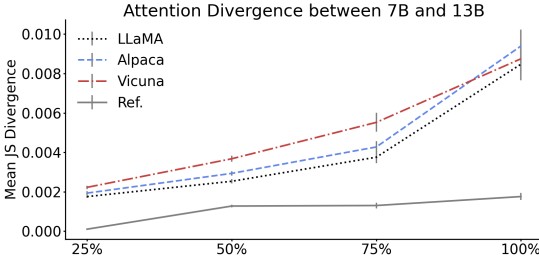

Figure 2: Mean J-S divergence between 7B and 13B model attention in each layer quarter, measured on non-instruction sentences.

### 5.1 General Attention Divergence

#### 5.1.1 Scaling Significantly Changes the General Attention

The general attention divergence between models in different sizes is shown in Figure 2. The results shows that LLaMA, Alpaca and Vicuna all reveal divergence significantly higher than the reference values when scaled from 7B to 13B. This means scaling causes large change in the overall attention distribution.

#### 5.1.2 Instruction Tuning Has Limited Effect on General Attention

The general attention divergence between pre-trained and instruction tuned models is shown in Figure 3. It can be observed that only Vicuna 13B brings divergence above the reference values, while all the others don't. The same result can also be observed on instruction-prefixed sentences. This suggests that instruction tuning can only bring limited change to the models' general attention distribution. This can also be verified by other results in the following parts.

#### 5.1.3 Instruction Tuning Enhances Sensitivity to Instructions

The divergence between the models' attention obtained on the original and instruction-attached sentences are shown in Figure 4. There are two major observations. Firstly, LLaMA, Alpaca and Vicuna all show divergence significantly above the reference values between the two types of sentences, especially in the shallower layers, which means they all have sensitivity to instructions. Secondly, while the divergence of LLaMA drops in the deeper layers, the divergence of Alpaca and Vicuna does

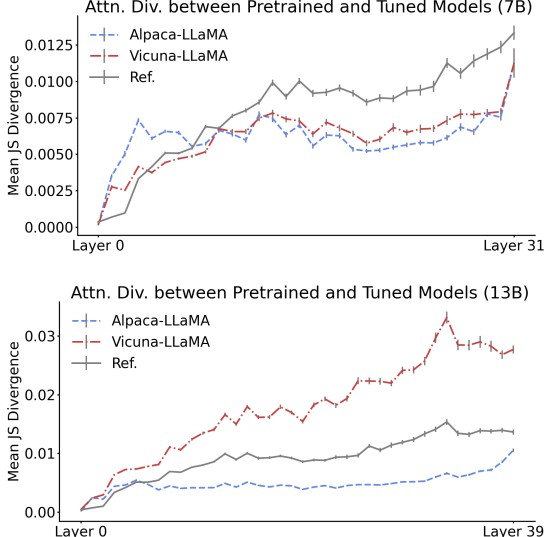

Figure 3: General attention divergence between pretrained and instruction-tuned models in 7B and 13B, measured on non-instruction sentences.

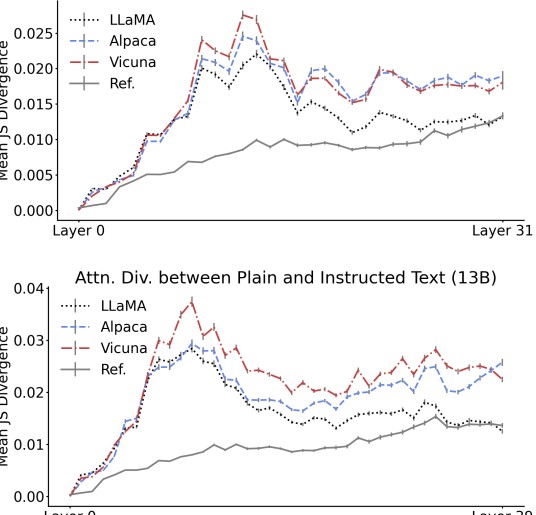

Figure 4: Attention divergence on the original and instruction-attached text of 7B and 13B models.

not go down, but rather up, suggesting higher sensitivity to instructions in the deeper layers. This effect is not observed in the noise-prefixed scenario (See Appendix A).

From these two observations, we can conclude that the pretrained LLaMA has already obtained some sensitivity to instructions, i.e. to change its attention mode when it meets instructions, and this sensitivity is further strengthened by instruction tuning, which makes this sensitivity higher in the deeper layers. Because in causal language models like LLaMA, the last few layers are related to generation, this effect of instruction tuning may bring more suitable generation for a certain instruction.

## 5.2 Human Resemblance

### 5.2.1 Higher Human Resemblance, Lower Prediction Loss

The per-token NTP loss and the maximum layerwise human resemblance scores are compared in Table 1 and Figure 5. One can tell from it that, within the considered range of parameter sizes, the human resemblance is negatively correlated with the NTP loss, where the resemblance goes higher nearly linearly with the loss going lower. The Pearson's correlation is $-0.875$ for L1 ($p < 0.002$) and $-0.917$ for L2 ($p < 0.0005$), supporting a significant and strong negative linear correlation.

This result shows that, the human resemblance we defined is positively related to the language modeling performance of the LLMs considered in

this research, giving a practical meaning to our human resemblance analysis. Also, based on this result, we know the pursuit for higher human resemblance is consistent with the target of training better LLMs. Similarly, factors negatively impacting the human resemblance could also harm the models' language modeling performance, such as instruction tuning brings lower human resemblance as well as higher NTP loss.

| Size | Name | Loss | L1(%) | L2(%) |
|------|------|------|-------|-------|
| 774M | GPT-2 | 0.3264 | 34.99 | 40.03 |
| | LLaMA | 0.2408 | 53.04 | 62.44 |
| 7B | Alpaca | 0.2646 | 52.51 | 61.71 |
| | Vicuna | 0.2593 | 51.90 | 61.19 |
| | LLaMA | 0.2406 | 55.66 | 64.20 |
| 13B | Alpaca | 0.2634 | 55.05 | 64.46 |
| | Vicuna | 0.2847 | 54.26 | 61.31 |
| 30B | LLaMA | 0.2372 | 63.16 | 69.40 |
| 65B | LLaMA | 0.2375 | 64.05 | 70.07 |

Table 1: Comparison between NTP loss and the human resemblance scores

### 5.2.2 Scaling Enhances Human Resemblance

The layerwise human resemblance of pretrained models in different scales are shown in Figure 6, where the LLaMAs are very different from GPT-2 Large in two ways. First, the layerwise human resemblance of the LLaMAs are much higher than GPT-2, but the gaps among the LLaMAs are small; Second, human resemblance of GPT-2 drop quickly in the deeper layers, while it remains high across all layers in the LLaMAs. one can refer from these

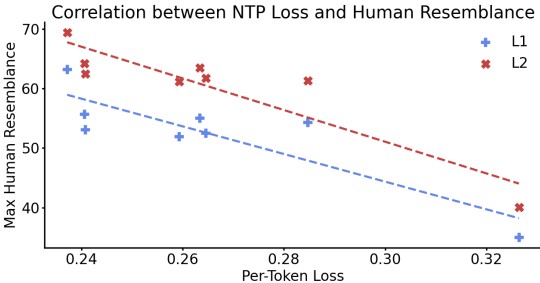

Figure 5: The correlation between the per-token negative log likelihood loss and the max layerwise human resemblance of LLMs.

results that LLaMA has undergone a phase change in terms of human-like attention from the level of GPT-2. However, the scaling from 7B to 30B does not cause another phase change.

Table 2 and Figure 7 show the max layerwise human resemblance increases linearly while the model parameter size increases exponentially, where the Pearson's correlation is 0.989 for L1 ($p < 0.002$) and 0.964 for L2 ($p < 0.01$). This agrees with the scaling law of LLMs (Henighan et al., 2020).

This result shows that, within the scope of this study, scaling significantly enhances the human resemblance. If the scaling law continues to take effect, the human resemblance is expected to reach 98.80% for L2 and 88.82% for L1 at the scale of 100B. However, because the law is expected to fail after reaching a threshold (Hestness et al., 2017), the relation between human resemblance and model scale is expected to change before reaching that scale.

| Model | | Resemblance (%) | |
|---|---|---|---|
| Name | Size | L1 | L2 |
| GPT-2 | 774M | 34.99 | 40.03 |
| | 7B | 53.04 | 62.44 |
| LLaMA | 13B | 55.56 | 64.20 |
| | 30B | 63.16 | 69.40 |
| | 65B | **64.05** | **70.07** |

Table 2: Max layerwise human resemblance of models ranging from 774M to 65B.

### 5.2.3 Instruction Tuning Harms Human Resemblance

Table 3 shows the max layerwise human resemblance of the pretrained and instruction tuned models. It shows that the instruction tuned Alpaca and Vicuna models have lower max layerwise human resemblance to both L1 and L2 than LLaMA in

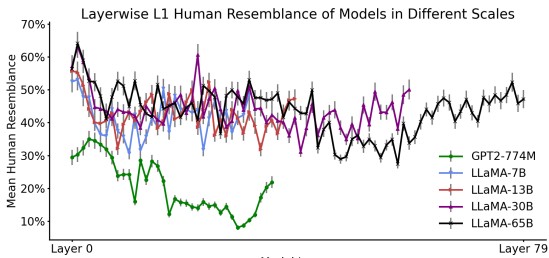

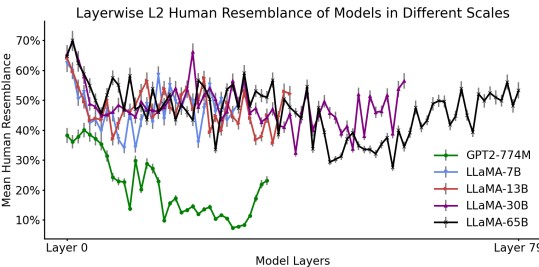

Figure 6: Layerwise L1 and L2 human resemblance for different sized pretrained models.

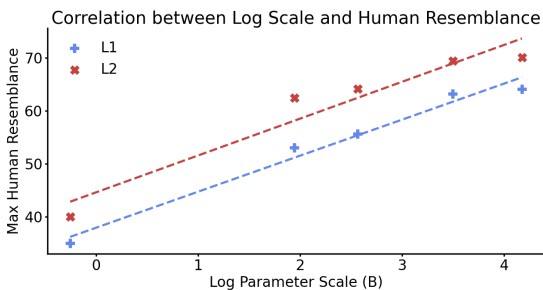

Figure 7: The change of max layerwise human resemblance caused by the scaling of model sizes, where the x-axis is log scaled.

both sizes, which suggests that instruction tuning causes decline in the human resemblance of attention. However, the decline is minor compared with the increase brought by scaling, which agrees with the limited effect of instruction tuning.

We also conduct relative t-tests to detect significant difference in layerwise human resemblance ($p = 0.05/n_{\text{test}}$). The numbers of significant layers are shown in Table 4, and the complete lists of layers are in Table 6, 7 in Appendix C). The tables show that instruction tuning enhances or reduces human resemblance on about the equal numbers of layers on the 7B models, but reduces human resemblance to much more layers on the 13B models. Also, the layers with significantly lower resemblance are distributed across all layers. This result means that, while the total effect of instruction tuning is small, the drop in human resemblance caused by it is significant and widespread.

To sum up, instruction tuning does small but significant harm to the human resemblance of LLMs, which in turn may bring potential damage to their language perception. This challenges the common assumption that instruction tuning can activate the model's ability to perceive language with human-aligned behaviors. However, this negative effect could be compensated by scaling, for scaling increases the human resemblance in a larger degree. This provides a novel reason for further scaling.

| Model | | Resemblance(%) | |
|---|---|---|---|
| Size | Name | L1 | L2 |
| | LLaMA | 53.04 | 62.44 |
| 7B | Alpaca | 52.51 | 61.71 |
| | Vicuna | 51.90 | 61.19 |
| | LLaMA | 55.66 | 64.20 |
| 13B | Alpaca | 55.05 | 63.46 |
| | Vicuna | 54.26 | 61.31 |

Table 3: Max layerwise human resemblance scores of pretrained and instruction tuned models. This shows instruction tuning causes small decline in the max layerwise human resemblance.

| Model | | Higher | | Lower | |
|---|---|---|---|---|---|
| Size | Name | L1 | L2 | L1 | L2 |
| 7B | Alpaca | 3 | 8 | 4 | 7 |
| | Vicuna | 9 | 10 | 8 | 9 |
| 13B | Alpaca | 4 | 7 | 16 | 25 |
| | Vicuna | 5 | 8 | 14 | 16 |

Table 4: Numbers of Alpaca and Vicuna layers with significantly higher or lower human resemblance compared with LLaMA in the same sizes.

### 5.2.4 All Models Show Higher L2 Resemblance

Comparing the L1 and L2 human resemblance of all models (LLaMA, Alpaca and Vicuna, 7B and 13B) in Table 2 and Table 3, an advantage of L2 over L1 can be observed steadily. Independent t-test also shows that, all models considered in this research show significantly higher resemblance to L2 than to L1 ($p = 0.05/n_{\text{test}}$ to avoid false positives). This means the models are closer to non-native English learners than to native English speakers in attention, though they are all trained mainly on English data. Furthermore, this trend is not reversed by the scaling or instruction tuning within the scope of this research, which suggests that it is a innate feature of LLMs. We will look further into this phenomenon in the next section.

### 5.3 Trivial Pattern Reliance

#### 5.3.1 L2 Relies More on Trivial Patterns

The results of LR scores between trivial patterns and L1 and L2 human saccade is shown in Table 5, where L2's score is higher than L1's in minimum, maximum and mean values. Independent t-test also supports a significant advantage of L2 over L1 on the regression scores ($p < 5 \times 10^{-8}$). This means one significant difference between L1 and L2 English speakers is that L2 people show more trivial and fixed attention mode than L1 people while reading, which suggests a weaker language understanding ability as those patterns contain no linguistic or factual information.

This finding can help us interpret the difference between the models' L1 and L2 human resemblance. Note that all models considered in this research are trained mainly on English, but show higher attention resemblance to L2 subjects, this result suggests that their language perception is not ideal.

| Group | Min | Max | Mean | SE |
|---|---|---|---|---|
| L1 | 0.0048 | 0.1525 | 0.0473 | 0.0037 |
| L2 | 0.0134 | 0.2165 | 0.0894 | 0.0060 |

Table 5: Regression scores of human saccade on the trivial patterns, across L1 and L2 subjects. SE stands for standard errors.

#### 5.3.2 Scaling Reduces Trivial Pattern Reliance

The relative difference between 13B and 7B models in trivial pattern reliance is shown in Figure 8, where the decline in the deeper layer quarters is substantial across all models. There is also a consistent increase in the second quarter, but the amplitude is small, so the total trivial pattern reliance is reduced. This phenomenon is also observed between LLaMA 13B and 30B (Figure 11 in Appendix B). This means scaling can effectively reduce the trivial pattern reliance, especially in the deeper layers, indicating a more contextualized and adaptive language perception.

#### 5.3.3 Instruction Tuning Increases Trivial Pattern Reliance

Figure 9 shows the relative difference between pretrained and instruction tuned models in trivial pattern reliance, where the tuned ones show reliance gain in the deeper layers. There is also a small drop in the second quarter, but the total trivial pattern reliance is increasing. Also, total increase here is

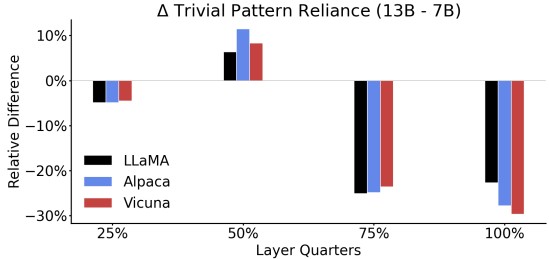

Figure 8: The difference between trivial pattern reliance of 7B and 13B model attention.

also smaller in amplitude than the reduction caused by scaling, again supporting the limit effect of instruction tuning.

Also, one can tell from Figure 8 and Figure 9 that scaling and instruction tuning changes the trivial pattern reliance in opposite directions. However, because the effect of instruction tuning is also smaller than that of scaling, as long as the model size continues to grow, this gain in trivial pattern reliance brought by instruction tuning could be compensated or even covered, offereing another reason for further scaling.

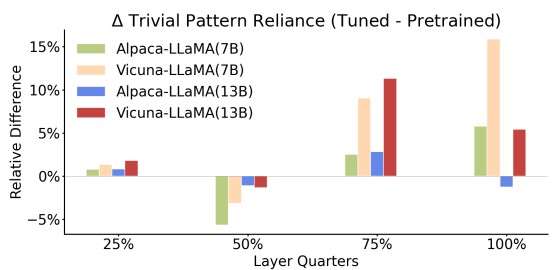

Figure 9: The difference in trivial pattern reliance of attentions in pretrained and instruction-tuned models.

## 6 Conclusion

This research evaluates the effects of scaling and instruction tuning on LLMs' attention. We find that scaling effectively changes the general attention distribution, enhances the human resemblance, and reduces the trivial pattern reliance, while instruction tuning does the opposite, but also increases the models' sensitivity to instructions. Furthermore, we find current open-sourced LLMs closer to non-native English speakers in language perception, which contains more trivial attention patterns. To the best of our knowledge, we are the first to analyze the effect of instruction tuning on the human resemblance of model attention. We hope this can inspire future work on analyzing LLMs.

## Limitations

One key limitation of this study is that we cannot apply our current method to closed-sourced LLMs such as ChatGPT, for their layerwise attention scores are unavailable. With the rapid development of open LLMs, it is hopeful that we can examine these findings on the largest and most advanced LLMs in the near future.

Besides, though we demonstrate the difference in LLMs' resemblance to L1 and L2 people, and partially explained the effect by trivial pattern reliance, it is not enough to explain the underlying cause or consequences of this phenomenon. We do have some observations, such as the inter-subject correlation in the L1 group ($0.1560 \pm 0.0073$) is lower than L2 ($0.2463 \pm 0.0090$), and L1 subjects tend to look back to pervious words less frequently than L2 subjects. However, the underlying mechanism is still unclear. We hope to dive deeper into this non-native resemblance in our following study.

Another limitation of this work comes from the way we obtain model attention. We used the attention scores when predicting the next single token, which makes sense. However, the attention patterns could be different in the following decoding steps. Also, it is possible that combining attention scores across multiple layers can improve the analysis. We will investigate this in our future work.

In addition, the Reading Brain dataset we used in this research is monolingual, which does not reflect the language perception process in the cross-lingual setting, which is also important for the application of LLMs. We also hope to examine our findings in the cross-lingual scenario in the future.

## Ethics Statement

The authors declare no competing interests. The human behavioral data we use is publicly available and does not contain personal information of the subjects.

## Acknowledgements

We would like to thank the anonymous reviewers for their insightful comments. Shujian Huang is the corresponding author. We thank Yunzhe Lv and Wenhao Zhu for finetuning the Alpaca 13B model. This work is supported by National Science Foundation of China (No. 62376116, 62176120), the Liaoning Provincial Research Foundation for Basic Research (No. 2022-KF-26-02).

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

## A   Attention divergence on plain and noise-prefixed text

As shown in Figure 10, the divergence between plain and noise-prefixed text of Alpaca and Vicuna is not significantly higher than LLaMA in most of the layers, which further supports our findings on the higher instruction sensitivity of instruction-tuned models.

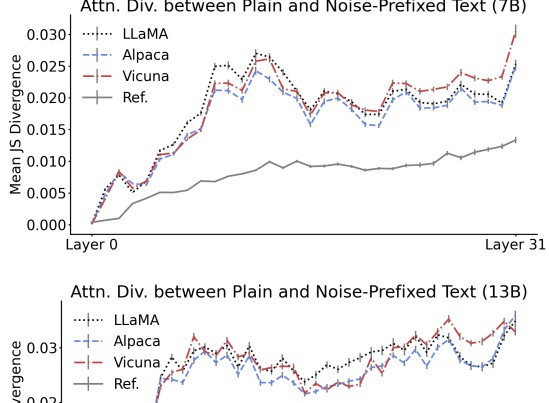

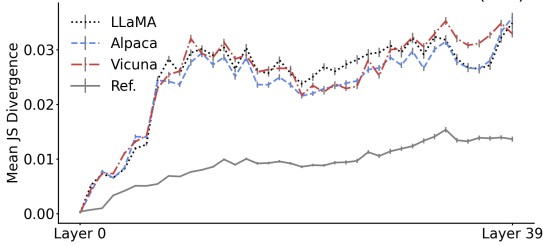

Figure 10: Attention divergence on the original and noise-attached text of 7B and 13B models.

## B   Layerwise trivial pattern reliance

Figure 11 shows the layerwise trivial pattern reliance of 7B and 13B models, as well as the results for LLaMA 30B. The reliance keeps decreasing as the model size scales up, especially in the deeper layers.

## C   Layers with significant changes in human resemblance after instruction tuning

Table 6 shows the list of layers in 7B Alpaca and Vicuna with significant change in human resemblance after instruction tuning compared with LLaMA, and Table 7 shows the list in 13B.

## D   Preliminary fMRI results

Some preliminary analysis on the fMRI data in the Reading Brain dataset is done, and the results com-

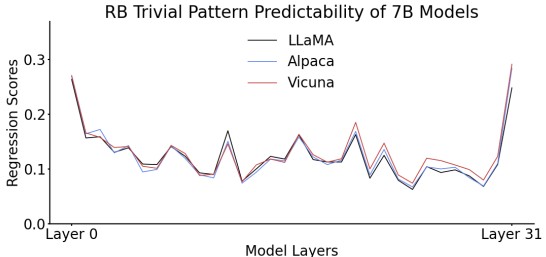

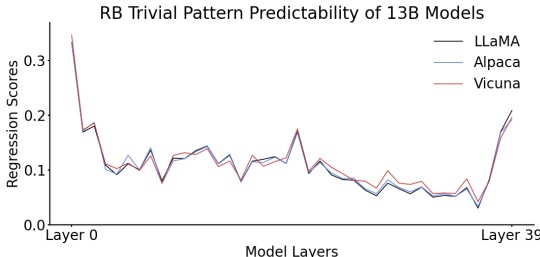

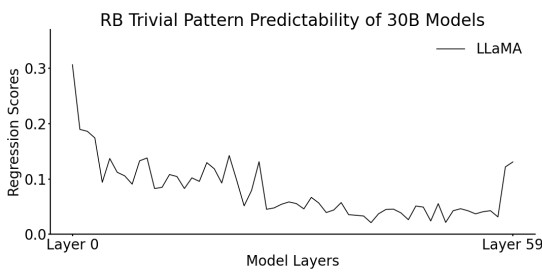

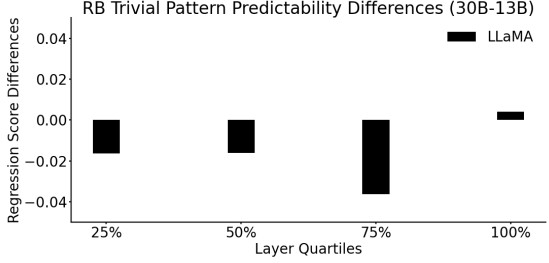

Figure 11: Lyerwise trivial pattern reliance of 7B and 13B models. "RB" stands for "Reading Brain".

| Relation | Group | Alpaca | Vicuna |
|---|---|---|---|
| Higher | L1 | 2, 5, 7 | 12, 13, 17, 18, 21, 26, 28, 29, 30 |
| | L2 | 2, 5, 6, 8, 12, 15, 21, 27 | 3, 13, 15, 18, 21, 23, 26, 28, 29, 30 |
| Lower | L1 | 4, 10, 23, 31 | 0, 1, 4, 5, 9, 11, 20, 31 |
| | L2 | 0, 10, 14, 19, 23, 28, 31 | 0, 1, 4, 5, 9, 11, 14, 20, 31 |

Table 6: Layers of Alpaca and Vicuna with significantly higher or lower human resemblance compared with LLaMA in 7B.

| Relation | Group | Alpaca | Vicuna |
|---|---|---|---|
| Higher | L1 | 16, 32, 33, 36 | 4, 16, 27, 33, 36 |
| | L2 | 1, 16, 19, 27, 32, 33, 36 | 4, 16, 19, 23, 27, 33, 36, 37 |
| Lower | L1 | 0, 2, 3, 5, 6, 7, 8, 9, 14, 18, 24, 25, 28, 30, 35, 39 | 0, 1, 3, 5, 6, 9, 10, 14, 17, 18, 24, 30, 34, 39 |
| | L2 | 0, 3, 4, 5, 6, 7, 8, 9, 10, 11, 12, 14, 15, 17, 18, 20, 24, 25, 28, 29, 30, 31, 35, 38, 39 | 0, 2, 3, 5, 6, 8, 9, 10, 11, 12, 14, 17, 18, 20, 21, 30 |

Table 7: Layers of Alpaca and Vicuna with significantly higher or lower human resemblance compared with LLaMA in 13B.

ply with our findings on saccade data. We investigate several linguistics-related brain regions, and find that the highest correlation of fMRI signals to model attention are from the anterior left temporal gyrus (Figure 12). The fMRI data from this region is pre-processed into matrix forms as saccade data. Figure 14 gives examples of individual and group mean fMRI matrices.

The findings are: (1) The models' NTP loss significantly ($p < 0.02$ for both L1 and L2) and strongly negatively ($r = -0.777$ for L1, $r = -0.796$ for L2) correlates with fMRI resemblance (Figure 13); (2) The models' highest layerwise fMRI resemblance significantly ($r > 0.99$, $p < 0.001$ for both L1 and L2) correlates with the model scale in logarithm (Figure 15, 16); (3) While scal-ing enhances human fMRI resemblance, instruction tuning does not (Table 8); (4) Resemblance to L2 fMRI is consistantly higher than that to L1.

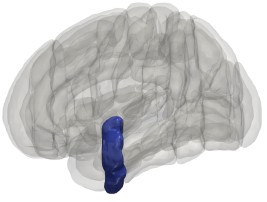

Figure 12: An illustration of the anterior left temporal gyrus in the human brain.

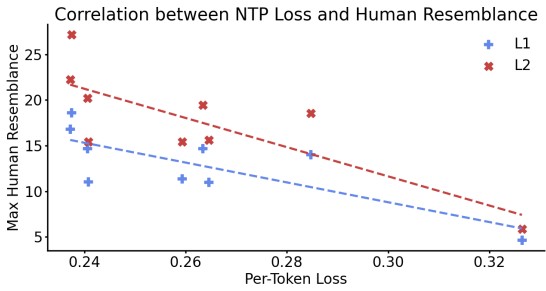

Figure 13: The correlation between per-token negative log likelihood loss and max layerwise fMRI resemblance of the LLMs.

| Model | | Resemblance(%) | |
|---|---|---|---|
| Size | Name | L1 | L2 |
| | LLaMA | 11.02 | 15.45 |
| 7B | Alpaca | 10.99 | 15.61 |
| | Vicuna | 11.35 | 15.44 |
| | LLaMA | 14.69 | 20.19 |
| 13B | Alpaca | 14.67 | 19.44 |
| | Vicuna | 14.02 | 18.57 |

Table 8: Max layerwise human fMRI resemblance scores of pretrained and instruction tuned models. This shows instruction tuning causes small decline in the max layerwise human resemblance.

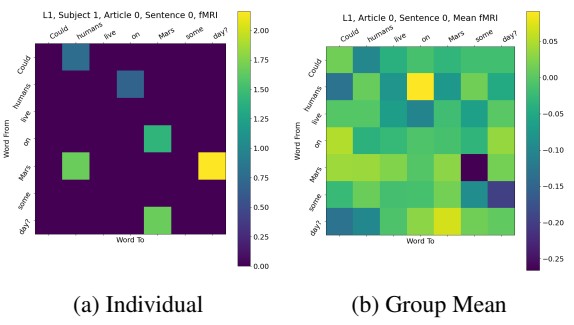

(a) Individual      (b) Group Mean

Figure 14: Examples of individual and group mean fMRI signals of the sentence "Could humans live on Mars some day?".

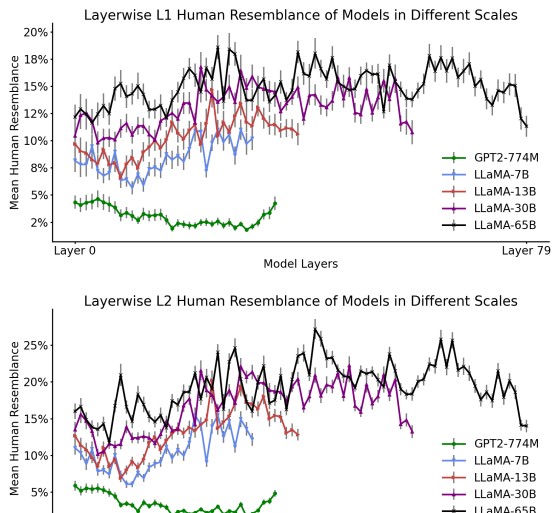

Figure 15: Layerwise L1 and L2 human fMRI resemblance for different sized pretrained models.

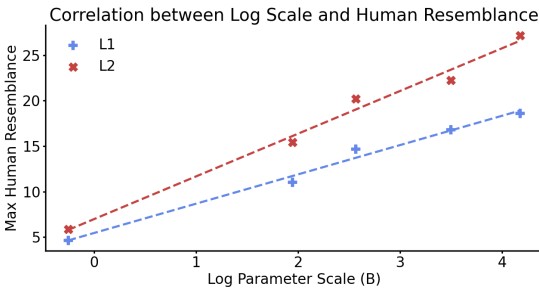

Figure 16: The change of max layerwise human fMRI resemblance caused by scaling. The x-axis is log scaled.