# OpenReview forum: "Roles of Scaling and Instruction Tuning in Language Perception: Model vs. Human Attention"
_EMNLP/2023/Conference — EMNLP 2023 Findings_

### Official Review · Reviewer_GxRX · 2023-07-28

**Soundness:** 2

**Excitement:**

2: Mediocre: This paper makes marginal contributions (vs non-contemporaneous work), so I would rather not see it in the conference.

**Missing References:**

There has been quite some work on the connection between model attention/attribution and human attention/reading importance via eye tracking experiments:
- "Relative Importance in Sentence Processing" - Beinborn & Hollenstein (2021)
- "A Cross-lingual Comparison of Human and Model Relative Word Importance" - Morger et al. (2022)
- See also the "CMCL 2021 Shared Task on Eye-Tracking Prediction" (https://aclanthology.org/2021.cmcl-1.7/)

The paper also misses a lot of papers that have recently presented better methods of interpreting attention by looking further then just the attention weights and incorporating the content of the value vectors as well.
- "Quantifying Attention Flow in Transformers" - Abnar & Zuidema (2020)
- "Attention is Not Only a Weight: Analyzing Transformers with Vector Norms" - Kobayashi et al. (2020)
- "Incorporating Residual and Normalization Layers into Analysis of Masked Language Models" - Kobayashi et al. (2021)
- "Measuring the Mixing of Contextual Information in the Transformer" - Ferrando et al. (2022)
- "Quantifying Context Mixing in Transformers" - Mohebbi et al. (2023)
- "Explaining How Transformers Use Context to Build Predictions" - Ferrando et al. (2023)

**Paper Topic And Main Contributions:**

The paper investigates the impact of 2 important factors in LLM research, **scaling** and **instruction tuning**, with respect to how they affect Transformer attention and align with 'human attention'. Using the Jensen-Shannon divergence as a metric for measuring the difference of attention distribution between models, and L1/L2 speakers, the authers uncover various surprising findings on the role of scaling and instruction tuning.

**Questions For The Authors:**

1. Regarding the experiment described at the end of Section 3.1: how do you account for the fact that the attention distribution will be different because it also flows to the tokens in the instruction itself?
2. Did you consider aggregating LM attention patterns across multiple layers? For LMs the attention is distributed across layers, so it is not completely 'fair' to compare an individual layer's attention to human attention as a whole.

**Reasons To Accept:**

- A lot of recent work on instruction tuning has focused on the quality and behavioural change of this procedure. The focus on change in model internals, as presented in this work, is an exciting and fruitful line of work, that will provide better insights into the reasons why instruction tuning is so successful.
- The authors present an extensive experimental setup that tackles various interesting questions regarding the impact of instruction tuning and scale.

**Reasons To Reject:**

- The paper does not seem to be aware of highly related work on the relation between human 'attention' (i.e. eye-tracking data) and LM 'attention'. In earlier work it has been shown that attribution methods yield a better fit to human attention than raw attention (Beinborn & Hollenstein, 2021), and the paper would benefit from engaging with this literature more (see below for references).
- The paper also does not seem to be aware of a recent line of work that devises better methodologies of interpreting LM attention (see below for references).
- The paper is too vague about "human attention" and would benefit from engaging more with literature from the cognitive science about attention in reading comprehension.
- The paper would benefit from a better restructuring/rewriting, there are a lot of unnecessary subsubsections and 1-sentence paragraphs.
- The paper would benefit from a more detailed analysis of the difference between L1 and L2 speakers, that reaches further than a stronger 'trivial pattern reliance' of L2 speakers.

**Reproducibility:**

3: Could reproduce the results with some difficulty. The settings of parameters are underspecified or subjectively determined; the training/evaluation data are not widely available.

**Reviewer Confidence:**

3: Pretty sure, but there's a chance I missed something. Although I have a good feel for this area in general, I did not carefully check the paper's details, e.g., the math, experimental design, or novelty.

**Typos Grammar Style And Presentation Improvements:**

## Suggestions
- Explain what types of "human attention" you use for your analysis, right now in Section 3 you have not explained this at all yet, and only in Section 4.2 this is briefly touched upon. This is important though, as you place a lot of importance on this "human attention" in your analysis, so it should be clear to a reader what exactly you mean by it.
- Refrain from using the term "prove" when there is no theoretical proof involved, you merely "demonstrate" something here.

## Style / Typos
- The sentence around L127 (On the other hand...) is hard to understand.

---

> ### Author Rebuttal · Authors · 2023-08-29
>
> Thank you for the very detailed and constructive comments. Below are our answers to the issues, questions and suggestions.
>
> ## Replies to the reasons for rejection
>
> > - The paper does not seem to be aware of highly related work on the relation between human 'attention' (i.e. eye-tracking data) and LM 'attention'. In earlier work it has been shown that attribution methods yield a better fit to human attention than raw attention (Beinborn & Hollenstein, 2021), and the paper would benefit from engaging with this literature more (see below for references).
>
> Thank you very much for the references. The papers are indeed highly related to our work, and we will add these citations in the revised manuscript. Attribution methods are good for explaining relative word importance (Beinborn & Hollenstein, 2021), but they are designed for showing the effect of the input on the output. Instead, we use attention analysis in this paper because the attention scores can show the interaction between the input tokens, resembling the form of human saccade matrices. Indeed, our results show that raw attention scores of recent LLMs such as LLaMA already correlate with human saccade patterns, suggesting that raw attention scores of large transformer models also contain useful information for interpretation as attribution methods do.
>
> > - The paper also does not seem to be aware of a recent line of work that devises better methodologies of interpreting LM attention (see below for references).
>
> The papers listed by the reviewers include several new and better methods (e.g., Attention Flow, ALTI, etc.) to interpret attention from transformers other than using the raw attention scores. We will add citations to them in the revised version. However, these attention interpretation methods also have some limitations. For example, ALTI are mainly designed for encoders, and Attention Flow averages over all attention heads to reduce calculation, which loses head-wise information. Instead, we think that raw attention already contains some basic patterns in language perception and is good enough for our analysis. In this regard, we have repeated the divergence analyses using Attention Rollout, and the results are consistent with our findings.
>
> > - The paper is too vague about "human attention" and would benefit from engaging more with literature from the cognitive science about attention in reading comprehension.
>
> We fully agree and will include more literature from cognitive science about human attention in reading comprehension, such as Scanpath (mentioned by reviewer #dzum), reading times and working memory for the background of our research.
>
> > - The paper would benefit from a better restructuring/rewriting, there are a lot of unnecessary subsubsections and 1-sentence paragraphs.
>
> Thank you for the very helpful suggestion. We will incorporate it in our revised paper.
>
> > - The paper would benefit from a more detailed analysis of the difference between L1 and L2 speakers, that reaches further than a stronger 'trivial pattern reliance' of L2 speakers.
>
> Thank you for your useful suggestion. We provide below some more analyses of the two language groups. The correlation between L1-average and L2-average saccade is 0.80, which suggests that they are different to some degree. The Inter-Subject Correlation is 0.1560(+-0.0073) for L1 and 0.2463(+-0.0090) for L2, suggesting that L1 individuals have more diverse reading patterns, which may also partly explain why the model fits better to L2.  Also, we observed different behavioral patterns of L1 and L2 subjects, such as L1 subjects tend to look back on previous words less than L2, but we are still searching for a good way to quantify them. We will add this information to the revised manuscript.
>
> ## Answers to the questions:
>
> > 1. Regarding the experiment described at the end of Section 3.1: how do you account for the fact that the attention distribution will be different because it also flows to the tokens in the instruction itself?
>
> Thank you for raising this issue. We agree that the prefix itself would affect the total attention distribution, and reviewer #pjsV also mentioned this issue and suggested comparing plain text and noise+plain text to account for the effect of extra prefix tokens. We have conducted the suggested analysis and the result shows that the divergence between plain and noise+plain text of Alpaca and Vicuna is not significantly higher than LLaMA in most of the layers, which further supports our findings. We will add this new result to the revised version.
>
> > 1. Did you consider aggregating LM attention patterns across multiple layers? For LMs the attention is distributed across layers, so it is not completely 'fair' to compare an individual layer's attention to human attention as a whole.
>
> Thank you for the insightful suggestion.  We agree that the model attention is distributed across layers. Our analysis of the layer-wise attention is based on the model mechanism that the representations in higher transformer layers contain more contextual information induced by previous layers, which would reflect the process of contextual language perception. However, we will add the results on attention across layers using methods such as Attention Rollout and Attention Flow to tackle this issue.
>
> ## Replies to the suggestions
>
> > - Explain what types of "human attention" you use for your analysis, right now in Section 3 you have not explained this at all yet, and only in Section 4.2 this is briefly touched upon. This is important though, as you place a lot of importance on this "human attention" in your analysis, so it should be clear to a reader what exactly you mean by it.
>
> Thank you for the useful suggestion. The "human attention" we refer to in this work is the human saccade patterns, which reflect the participation of contextual words when the subject is perceiving the current word. We will make this clearer in the introduction part of the revised version.
>
> > - Refrain from using the term "prove" when there is no theoretical proof involved, you merely "demonstrate" something here.
>
> Thank you for pointing out this issue. We will substitute the use of "prove" with more appropriate words such as "show" or "demonstrate" in the revised version.
>
> ## Style / Typos
>
> > - The sentence around L127 (On the other hand...) is hard to understand.
>
> Thank you very much for pointing this out. The correct sentence should be "On the other hand, we select several trivial attention patterns which correspond to meaningless structures as reference for bad attention, and design the metric of trivial pattern reliance for any given attention." We will correct this sentence in our revised version.
>
> ## Response to the reproducibility score
>
> We will make parameters and other settings clearer in the next version. All the model weights, codes and intermediate results will also be publicly available on GitHub after acceptance.

---

### Official Review · Reviewer_dzum · 2023-08-04

**Soundness:** 2

**Excitement:**

2: Mediocre: This paper makes marginal contributions (vs non-contemporaneous work), so I would rather not see it in the conference.

**Missing References:**

@article{von2013scanpaths,
  title={Scanpaths reveal syntactic underspecification and reanalysis strategies},
  author={Von der Malsburg, Titus and Vasishth, Shravan},
  journal={Language and Cognitive Processes},
  volume={28},
  number={10},
  pages={1545--1578},
  year={2013},
  publisher={Taylor \& Francis}
}

@article{von2015determinants,
  title={Determinants of scanpath regularity in reading},
  author={von der Malsburg, Titus and Kliegl, Reinhold and Vasishth, Shravan},
  journal={Cognitive science},
  volume={39},
  number={7},
  pages={1675--1703},
  year={2015},
  publisher={Wiley Online Library}
}

@article{oh2023does,
  title={Why does surprisal from larger transformer-based language models provide a poorer fit to human reading times?},
  author={Oh, Byung-Doh and Schuler, William},
  journal={Transactions of the Association for Computational Linguistics},
  volume={11},
  pages={336--350},
  year={2023},
  publisher={MIT Press}
}

@misc{oh2023transformerbased,
      title={Transformer-Based LM Surprisal Predicts Human Reading Times Best with About Two Billion Training Tokens},
      author={Byung-Doh Oh and William Schuler},
      year={2023},
      eprint={2304.11389},
      archivePrefix={arXiv},
      primaryClass={cs.CL}
}

**Paper Topic And Main Contributions:**

This paper tests the relative importance of model size ("scaling") and model objective (instruction-tuned or not) in the fit of attention patterns in autoregressive (next-word prediction) large language models to "human attention." This is quantified by comparing the self-attention patterns in the models to co-occurrences between saccades in human eyetracking data for both first language and second language English sentences. The authors compare several models (Alpaca, LLAMA, and Vicuna) at different model sizes and analyze according to model layer depth as well. Generally speaking, they find that size is more important for matching human attention patterns than instruction training, that the models seem to resemble L2 readers more than L1 readers, and that instruction training weakens the resemblance to human attention patterns. They also present a baseline for degree of similarity that is meant to be trivial.

ETA: I have read the author response and appreciate the time the authors put in to making a response to my concerns. I will keep my score the same.

**Reasons To Accept:**

* The paper uses an uncommon dataset (the Reading Brain data) for evaluation
* The paper assesses new, cutting edge models that are available in an open capacity (i.e., not closed source)
* The experiments are extensive and seem appropriate for the authors' research questions

**Reasons To Reject:**

* I think the paper is too atheoretical. I do not see a clear motivation for human evaluation -- it is sort of presented as-is. I do not think that this human evaluation helps us to understand how these models differ from each other than any other dataset (that is, not this dataset of reading times, or any dataset of reading times) would do.
* I do think that looking at eyetracking saccade co-occurrences is interesting but this approach has been done in the literature before -- scanpath work should be cited to motivate the approach since it is not novel (though it is an interesting way of capturing something)
* I would not refer to these saccade data as "human attention" -- I would steer clear of calling *anything* _human attention_ without very carefully situating the analyses in over a century of psychophysical literature and being extremely clear about what cognitive processes are going on. The model "attention" is also not "attention" in the same way we talk about it colloquially -- something I suspect the authors know.
* I find the focus on L1/L2 also perplexing. The authors should be extremely clear about what they think differentiates the L1/L2 data and why this is relevant to LLMs. As it is, I think that the authors frame L2 readers as having a kind of "deficit" in processing that is not necessarily appropriate. Rather, L2 reading may simply engage different kinds of cognitive skills that L1 readers may use less or not at all.
* The authors repeatedly state that they have "proved" things throughout the manuscript and that is simply not how scientific inquiry goes.
* Despite the novelty of the evaluation dataset, the analyses are based on only ~1500 words from ~150 sentences. It is extremely challenging to get enough data to overcome the sparsity issue for human eye movements as well as for model outputs -- what can we infer from 150 sentences? It would be nice to see this done with other eyetracking corpora (e.g., Dundee, Provo, etc.) and I see no particular reason why the authors would only work with the dataset they used here.
* The authors should be aware of other work that has looked at training dataset size, e.g., Oh and Schuler (2023), both the TACL proceedings and the arxiv paper
* The authors should ideally also relate the work to some understanding of human language processing. I don't know what the connection is but using human psycholinguistic data solely for evaluation is really limiting given the massive industry that is correlational studies to human reading dynamics and LLMs.

**Reproducibility:**

3: Could reproduce the results with some difficulty. The settings of parameters are underspecified or subjectively determined; the training/evaluation data are not widely available.

**Reviewer Confidence:**

5: Positive that my evaluation is correct. I read the paper very carefully and I am very familiar with related work.

---

> ### Author Rebuttal · Authors · 2023-08-29
>
> Thank you for the very useful and constructive comments. Below are our answers to the issues and suggestions.
>
> ## Replies to the reasons for rejection
>
> > - I think the paper is too atheoretical. I do not see a clear motivation for human evaluation -- it is sort of presented as-is. I do not think that this human evaluation helps us to understand how these models differ from each other than any other dataset (that is, not this dataset of reading times, or any dataset of reading times) would do.
>
> We agree that our motivation may not be clearly conveyed in the manuscript, and we will make it clearer in the revised version. Our motivation for using human behavioral data for model evaluation stems from prior work that compares language models with human neuroimaging data during language comprehension (e.g., Hasson et al., 2020; Schrimpf et al., 2021). These studies suggest that models with higher human resemblance also perform better in NLP tasks. However, given recent breakthroughs in LLMs such as ChatGPT and LLaMA, it remains to be tested whether these newest models, with their larger scale and extra fine-tuning techniques, still align with human perception data.
>
> Our human evaluation provides new insights into the effects of scaling and instruction tuning, revealing interesting results (e.g., while instruction tuning is widely used to improve the overall performance of LLMs, it may not enhance human resemblance, whereas further scaling does). These insights cannot be obtained solely through NLP task scores. Additionally, our methods can be applied not only to the Reading Brain dataset but also to other datasets that capture human saccade behavior. We chose the Reading Brain dataset because it simultaneously records eye-tracking and fMRI data. We already have some preliminary results on the fMRI part and plan to further investigate it in the future.
>
> > - I do think that looking at eye-tracking saccade co-occurrences is interesting but this approach has been done in the literature before -- scanpath work should be cited to motivate the approach since it is not novel (though it is an interesting way of capturing something)
>
> Thank you for pointing out the inspiring work on Scanpath. These studies reveal the reading characteristics of different groups of people using the trajectories of eye-movement, which we think will improve our theoretical basis. We will add these references to the revised version. Also, we are not stating that we are the first to analyze the human saccade data. Instead, the novelty of our work mainly lies in the comparison between saccade data and model self-attention, which is especially affected by scaling and instruction-tuning.
>
> > - I would not refer to these saccade data as "human attention" -- I would steer clear of calling *anything* *human attention* without very carefully situating the analyses in over a century of psychophysical literature and being extremely clear about what cognitive processes are going on. The model "attention" is also not "attention" in the same way we talk about it colloquially -- something I suspect the authors know.
>
> We recognize that the term "human attention" may be too vague, and we will substitute it with more suitable terms such as "human saccade patterns" in the revised version. We are aware that the self-attention weights in transformer models are based on query-key-value matrix multiplications and softmax calculations, and are not the same as the "attention" in colloquial usage. However, from a higher conceptual level, the self-attention mechanism represents the reliance on contextual information, which also included in the human reading behavior. We believe this provides the ground for our comparison analyses.
>
> > - I find the focus on L1/L2 also perplexing. The authors should be extremely clear about what they think differentiates the L1/L2 data and why this is relevant to LLMs. As it is, I think that the authors frame L2 readers as having a kind of "deficit" in processing that is not necessarily appropriate. Rather, L2 reading may simply engage different kinds of cognitive skills that L1 readers may use less or not at all.
>
> We agree that this is a concern. However, we have stated in our paper that the LLMs employed in this research are all mainly trained (both pre-trained and instruction-tuned) on English data, so one can expect that they are closer to native speakers of English. However, our results showed that the models are actually closer to the L2 speakers. This is why we consider the models to be “sub-optimal”. We have never referred to L2 speakers as "deficit" readers. We merely stated that their saccade patterns are different from those of the L1 speakers, e.g. in trivial-pattern reliance.
>
> > - The authors repeatedly state that they have "proved" things throughout the manuscript and that is simply not how scientific inquiry goes.
>
> Thank you for pointing this out. We will substitute the use of "prove" with more appropriate words such as "show" or "demonstrate" in the revised version.
>
> > - Despite the novelty of the evaluation dataset, the analyses are based on only ~1500 words from ~150 sentences. It is extremely challenging to get enough data to overcome the sparsity issue for human eye movements as well as for model outputs -- what can we infer from 150 sentences? It would be nice to see this done with other eye-tracking corpora (e.g., Dundee, Provo, etc.) and I see no particular reason why the authors would only work with the dataset they used here.
>
> Thank you for the comment. We agree that the corpus size in the Reading Brain dataset is relatively small compared to other commonly used eye-tracking datasets such as Dundee. However, it is comparable to those used in previous work (Schrimpf et al., 2021) on comparing language models and human data such as fMRI (Pereira et al., 2018; Blank et al., 2014) and ECoG (Fedorenko et al., 2016). Those datasets also contain only about a hundred sentences. We use the Reading Brain dataset because it has simultaneously recorded eye-tracking and fMRI data. We already have some preliminary results on the fMRI part, and plan to work on it further in the future. We will also include supplementary analysis based on larger eye-tracking datasets to further support our findings in the revised version.
>
> > - The authors should be aware of other work that has looked at training dataset size, e.g., Oh and Schuler (2023), both the TACL proceedings and the arxiv paper
>
> We thank the reviewer for pointing out these two exciting papers. These studies give insights about how and why the surprisal derived from larger LMs is less correlated with the human reading time. Although the focus of our paper is model attention (an internal state) instead of surprisal (a reflection of the final prediction), we agree that these two papers are related to the analysis of human resemblance and scaling, and we will add these references in our revised paper.
>
> > - The authors should ideally also relate the work to some understanding of human language processing. I don't know what the connection is but using human psycholinguistic data solely for evaluation is really limiting given the massive industry that is correlational studies to human reading dynamics and LLMs.
>
> Thank you for the helpful suggestion. We believe that comparing human look-back saccades with model attention is indeed based on the reading dynamics behind them. For example, prior studies have found that look-backs in reading are related to working memory efficiency (Walczyk & Taylor, 1996), and that scanpaths reflect parsing strategies (von der Malsburg & Vasishth, 2013). We will make this link clearer in the revised version. Additionally, beyond correlational studies, our long-term goal is to extract and compare the reading mechanisms and dynamics of humans and machines at a higher level, and try to improve machine reading by incorporating human-like dynamics.
>
> ## Response to the reproducibility score
>
> We do not understand why the reviewer thinks our results cannot be replicated. We will make parameters and other settings clearer in the next version. All the model weights, codes and intermediate results will also be publicly available on GitHub after acceptance. We would also be very happy to invite the reviewer to check whatever results that they think cannot be replicated.

---

### Official Review · Reviewer_pjsV · 2023-08-05

**Soundness:** 4

**Excitement:**

4: Strong: This paper deepens the understanding of some phenomenon or lowers the barriers to an existing research direction.

**Missing References:**

N/A

**Paper Topic And Main Contributions:**

The paper investigates how scaling and instruction turning of LMs (mainly LLMs) impacts the self-attention distributions. It compares these to human attention based on a publicly available dataset built from recording human eye tracking whilst reading sentences.
The main conclusions are interesting, namely that scaling up LMs results in producing attention distributions closer to the human ones, and that instruction tuning the models actually reduces this similarity.
The paper is well written, contains a good amount of evidence for the claims made.

**Questions For The Authors:**

* The tokens in the instruction influence the attention over the tokens of interest. When you report results in Figure 4, is it based on renormalising the instruction+plain text attentions over only the tokens in the plain text?
* What are the attentions like between plain text and noise + plain text? Would this say anything about how the extra pre-pended instruction text impacts attentions?
* Also on Fig4, the divergence patterns in Figure4 are interesting, in that I have no good hypothesis as to why the max difference is about 1/3 into the model.
* Each result is based on one model only. Do you think retraining the model (same size, same data, same curriculum) would result in pretty much the exact same attentions?
* without having read the Clark and Manning papers (line 312), is there any other good options for this operation of squishing the token level attention matrix down to same dimensions as the word level one? Or is this used procedure the only real option?

**Reasons To Accept:**

It reports an interesting finding, namely that the self-attention distribution of transformer language models is not improved by instruction tuning, is improved by scaling, and most interestingly is closer to non-native speaker attention distributions than native speakers. These results are surprising  since instruction tuning results in (certain types) of improved performance and generalisation, whilst the greater similarity to non-native speakers hints at being able to focus on the self-attention mechanism to improve models further.
Several experiments reported to provide evidence for the claims made. These are done on public data and with public models.
This is also an important research area, as we have made great steps recently in model ability, but understanding how these models perform so well is less advanced.

**Reasons To Reject:**

No significant reason.

**Reproducibility:**

3: Could reproduce the results with some difficulty. The settings of parameters are underspecified or subjectively determined; the training/evaluation data are not widely available.

**Reviewer Confidence:**

3: Pretty sure, but there's a chance I missed something. Although I have a good feel for this area in general, I did not carefully check the paper's details, e.g., the math, experimental design, or novelty.

**Typos Grammar Style And Presentation Improvements:**

* Can you give equivalent numbers on the 7B model at line 289/290? Saves reader digging those up and would better justify your point that the 13B model you trained is sound.

---

> ### Author Rebuttal · Authors · 2023-08-29
>
> Thank you very much for the positive comments. Below are our answers to the questions and suggestions.
>
> ## Answers to the questions:
>
> > - The tokens in the instruction influence the attention over the tokens of interest. When you report results in Figure 4, is it based on renormalising the instruction+plain text attentions over only the tokens in the plain text?
>
> Thank you for raising this question. The J-S divergence reported in Figure 4 is based on renormalising the instruction+plain text attentions over only the tokens in the plain text. We will clarify this in the revised version of the manuscript.
>
> > - What are the attentions like between plain text and noise + plain text? Would this say anything about how the extra pre-pended instruction text impacts attentions?
>
> Thank you for the constructive suggestion. We have now tested this noise+plain text scenarios on the 7B models, and found that the divergence between plain and noise+plain text (the noise consists of 5 randomly generated English words) of Alpaca and Vicuna is not significantly higher than LLaMA in most of the layers, which further supports our findings on the higher instruction sensitivity of instruction-tuned models. We will add this new result to the revised version.
>
> > - Also on Fig4, the divergence patterns in Figure4 are interesting, in that I have no good hypothesis as to why the max difference is about 1/3 into the model.
>
> We agree that this is an intriguing phenomenon and needs further explanation. Since the first 1/3 of all layers are relatively shallow, we think that higher difference in these layers may indicate that, the instructions will alter the model perception at lower levels. The difference fades in the pre-trained-only model (LLaMA) in the deeper layers, but retains in the instruction tuned models (Alpaca, Vicuna). Further experiments will be needed to confirm our hypotheses.
>
> > - Each result is based on one model only. Do you think retraining the model (same size, same data, same curriculum) would result in pretty much the exact same attentions?
>
> Although the initial states,  random seed, etc., will affect the attention patterns of the LLMs to some extent, we do not think this will make too much difference to the result of the analyses. As pointed out by prior studies, some of the attention heads are indispensable for the correct prediction of transformer models (Viota, et al., 2019), so the attention patterns carried by these heads will be replicated in a successfully re-pretrained LM, in some of the layers and attention heads. And since our human resemblance calculation is based on regression over all attention heads and maximum over all layers, we expect the shared attention patterns to be located, and the human resemblance score to stay approximately the same.
>
> > - without having read the Clark and Manning papers (line 312), is there any other good options for this operation of squishing the token level attention matrix down to same dimensions as the word level one? Or is this used procedure the only real option?
>
> Thank you for pointing out this issue. There could be other ways to do this, such as preserving only the first token in each word (which loses information in other sub-word tokens); or aggregating the sub-word tokens based on their frequency or structural importance (which brings more computational complexity). Instead, we follow Clark and Manning’s method because we think it is reasonable. Viewing words as groups of tokens, the sum over "to" tokens (columns in the attention matrix)  and the average over "from" tokens (rows in the attention matrix) means we aggregate the attention scores between the context and all tokens in that group, while keeping the sum of attention scores in each row as 1.
>
> ## Presentation improvements:
>
> > - Can you give equivalent numbers on the 7B model at line 289/290? Saves reader digging those up and would better justify your point that the 13B model you trained is sound.
>
> Thank you for the suggestion. The MMLU score for the Stanford Alpaca 7B model is 40.93 for zero-shot and 39.18 for one-shot. (The zero-shot score is higher than the one-shot score, which could be due to longer context, and this is not observed in LLaMA-7B or Alpaca-13B.)  We will add this information in our revised version.
>
> ## Response to the reproducibility score
>
> We will make parameters and other settings clearer in the next version. All the model weights, codes and intermediate results will also be publicly available on GitHub after acceptance.

---

### Meta-Review · Area_Chair_R6NK · 2023-09-28

**Recommendation:** 3

**Metareview:**

The work is about the effects of scaling and instruction tuning on the attention distribution and language modeling performance of large language models (LLMs). The authors use two types of instructions: L1 for general language modeling and L2 for specific tasks. They measure the attention distribution using three metrics: entropy, human resemblance, and trivial attention patterns. They also compare the perplexity scores of different models on various datasets.

Pros:
1: An interesting and surprising finding that challenges the common assumption that instruction tuning improves the self-attention distribution of transformer language models.
2: Revealed a novel and intriguing phenomenon that the self-attention distribution of large language models is closer to non-native speaker attention distributions than native speakers, which opens up new avenues for research and improvement.

Cons:
1: Uses the term “human attention” loosely and without proper definition or justification, and does not relate it to the model attention or the cognitive processes involved in reading comprehension.
2: Didn't provide a detailed or meaningful analysis of the difference between L1 and L2 speakers, and only mentions a stronger trivial pattern reliance of L2 speakers, which does not explain the underlying causes or consequences of this phenomenon.

---

### Decision · Program_Chairs · 2023-10-07

**Decision:**

Accept-Findings

**Comment:**

The work is about the effects of scaling and instruction tuning on the attention distribution and language modeling performance of large language models (LLMs). The authors use two types of instructions: L1 for general language modeling and L2 for specific tasks. They measure the attention distribution using three metrics: entropy, human resemblance, and trivial attention patterns. They also compare the perplexity scores of different models on various datasets.

Pros:
1: An interesting and surprising finding that challenges the common assumption that instruction tuning improves the self-attention distribution of transformer language models.
2: Revealed a novel and intriguing phenomenon that the self-attention distribution of large language models is closer to non-native speaker attention distributions than native speakers, which opens up new avenues for research and improvement.

Cons:
1: Uses the term “human attention” loosely and without proper definition or justification, and does not relate it to the model attention or the cognitive processes involved in reading comprehension.
2: Didn't provide a detailed or meaningful analysis of the difference between L1 and L2 speakers, and only mentions a stronger trivial pattern reliance of L2 speakers, which does not explain the underlying causes or consequences of this phenomenon.